# *Pontoscolex corethrurus*: A homeless invasive tropical earthworm?

Angel I. Ortíz-Ceballos[1]*, Diana Ortiz-Gamino[1], Antonio Andrade-Torres[1], Paulino Pérez-Rodríguez[2], Maurilio López-Ortega[1]

1 Instituto de Biotecnología y Ecología Aplicada (INBIOTECA), Universidad Veracruzana, Col. Emiliano Zapata, Xalapa, Veracruz, México, 2 Programa de Estadística, Campus Montecillo, Colegio de Postgraduados, Montecillo, Estado de México, México

* angortiz@uv.mx

**Data Availability Statement:** All relevant data are within the manuscript and its Supporting Information files.

## Abstract

The presence of earthworm species in crop fields is as old as agriculture itself. The earthworms *Pontoscolex corethrurus* (invasive) and *Balanteodrilus pearsei* (native) are associated with the emergence of agriculture and sedentism in the region Amazon and Maya, respectively. Both species have shifted their preference from their natural habitat to the cropland niche. They contrast in terms of intensification of agricultural land use (anthropic impact to the symbiotic soil microbiome). *P. corethrurus* inhabits conventional agroecosystems, while *B. pearsei* thrives in traditional agroecosystems, i.e., *P. corethrurus* has not yet been recorded in soils where *B. pearsei* dwells. The demographic behavior of these two earthworm species was assessed in the laboratory over 100 days, according to their origin (OE; *P. corethrurus* and *B. pearsei*) food quality (FQ; soil only, maize stubble, *Mucuna pruriens*), and soil moisture (SM; 25, 33, 42%). The results showed that OE, FQ, SM, and the OE x FQ interaction were highly significant for the survival, growth, and reproduction of earthworms. *P. corethrurus* showed a lower survival rate (> mortality). *P. corethrurus* survivors fed a diet of low-to-intermediate nutritional quality (soil and stubble maize, respectively) showed a greater capacity to grow and reproduce; however, it was surpassed by the native earthworm when fed a high-quality diet (*M. pruriens*). Besides, *P. corethrurus* displayed a low cocoon hatching (emergence of juveniles). These results suggest that the presence of the invasive species was associated with a negative interaction with the soil microbiota where the native species dwells, and with the absence of natural mutualistic bacteria (gut, nephridia, and cocoons). These results are consistent with the absence of *P. corethrurus* in milpa and pasture-type agricultural niches managed by peasants (agroecologists) to grow food regularly through biological soil management. Results reported here suggest that *P. corethrurus* is an invasive species that is neither wild nor domesticated, that is, its eco-evolutionary phylogeny needs to be derived based on its symbionts.

**Funding:** The authors received no specific funding for this work.

**Competing interests:** The authors have declared that no competing interests exist.

## Introduction

Although humans have produced novel niches prior to the advent of agriculture, the innovation of domestication led to changes in the life cycle of one or a few species, and the local microenvironments were manipulated, especially soil biota [1–4]. The artificial landscapes that resulted from these practices (anthropocentric ecology) were exported as agricultural packages from the centers of origin [1, 2, 4]. Thus, over a relatively short period in the history of mankind, the expansion of agriculture has brought about the remodeling of biodiversity as one of the most significant anthropogenic impacts on terrestrial ecosystems [1, 2, 5].

Agriculture has given rise to uniform and predictable disturbed ecological niches (invasible habitats), which have proven highly beneficial for non-domesticated species or weeds [1, 2, 6], and some earthworm species. Blakemore [7] has suggested that the origins of cosmopolitan (invasive) earthworms at family level are associated with domestication centers of plants and animals; that is, the presence of earthworms in crop fields is as old as agriculture itself [7–9]. The terms of the Millennium Ecosystem Assessment highlight the catalytic role of earthworms regarding two environmental services [10], namely the formation of soil and biogeochemical cycles, both of which are prerequisites for other environmental services [10–11].

Most of the studies focused on earthworms have used species adapted to crops, and most of them are currently considered as invasive [11]. It has been documented that 3% of the diversity of earthworms are invasive species [12]. As an example, European earthworms are frequently mentioned as the main cause of an irreversible change in the diversity and functioning of ecosystems in North America (Wisconsin glaciation areas) that were previously free from earthworms 12 thousand years ago [13–15]. However, there is a deeply rooted positive attitude toward earthworms in human populations in North America, acknowledging their beneficial effects on agricultural soils and urban gardens [10, 16].

Among the invasive tropical earthworms, the endogeic species *Pontoscolex corethrurus* was collected and described in crop fields in Blumenau, Brazil 160 years ago [17–18]; it has a broad distribution range and is the most studied tropical species [19–20]. Native species also move across a region in a similar way to invasive species, in addition to natural displacements [5, 7, 21]. The native endogeic earthworm *Balanteodrilus pearsei* was first collected and described from Gongora cave in Okcutzcab, Yucatan 81 years ago [22]; it is distributed in the east and southeast of Mexico and Belize [19]; it dwells in natural and agricultural environments and is the most studied species native to Mexico. Most studies conducted with both species point to a positive influence of their biological activity on soil [20, 23], i.e., they do not meet the definition of pest [24]. For this reason, we use the term *invasive* with reference to the biogeographical status of the species, regardless of its impact on soil [24–25].

Similar to weeds [6, 8, 9, 26], it can be suggested that *P. corethrurus* and *B. pearsei* have shifted their preference from their natural habitat to agricultural environments, spreading geographically beyond their place of origin, and are currently key elements of agricultural environments. The presence of *P. corethrurus* and *B. pearsei* is associated with the development of pre-Columbian cultivation techniques in the Amazon [2, 27, 28, 29] and Maya [2, 30, 31] regions, respectively. For example, it is believed that *P. corethrurus* facilitated the formation of fertile soils in the Amazon area named "Terra Preta do Indo" [32, 33, 34, 35, 36, 37]. Both species have adapted to niches that emerged from agriculture [38], but contrast regarding the intensification of agricultural land use and/or the diversion of each from natural habitats (anthropic manipulation of soil). *P. corethrurus* is commonly found in conventional agrecosystems (use of fertilizers, herbicides, pesticides, and tillage), as well as in industrial (polluted with heavy metals, petroleum hydrocarbons, and others) and urban areas [20, 39, 40, 41]. *B. pearsei* inhabits soils managed under an agroecological approach (little human impact of the soil

microbiome), such as traditional agroecosystems (no use of industrial inputs) and in natural ecosystems [40, 42, 43]. *P. corethrurus* has been found coexisting with native species in some agroecosystems [41, 44, 45], but there are no records of its coexistence with *B. pearsei* so far [40, 42, 43].

A previous study of coexistence under controlled conditions showed no competitive interaction between *P. corethrurus* and *B. pearsei*, i.e., both can coexist [23]. However, the question to address is, why *P. corethrurus* has not invaded the agroecological niche of *B. pearsei*? Therefore, this work compared the demographic behavior of *P. corethrurus* vs. *B. pearsei* assuming that the survival rate of the invasive species decreases in soil populated by the native species.

## Materials and methods

### Ethics statement

No permits were required for the collection and laboratory trials. Soil and earthworms were provided by farmers with free of charge. The experimental procedure used in this study is detailed elsewhere [23].

### Soil

Soil was collected from a maize field (MM) rotated with the tropical legume velvet bean (*Mucuna prurien var. utilis*) located near the village Tamulté de las Sabanas (18˚08´N, 92˚47´W), 30 km east of Villahermosa, Tabasco, Mexico. The silty clay loam soil (41.5% sand; 26.8% clay; 31.6% silt) was air-dried in the shade at room temperature and sieved through a 2 mm mesh. The main chemical characteristics of this soil were: 2.7% organic matter; 0.14% total N; 11.4 C/N; pH ($H_2O$) of 6.3.

### Earthworms

Two tropical endogeic earthworm species were used in this study: *B. pearsei* (native) and *P. corethrurus* (invasive). *B. pearsei* was collected from the MM field, whereas *P. corethrurus* was collected from pastures at Huimanguillo (79 km southwest Villahermosa, 17˚48´N, 93˚28´W), given its absence in the former site. All earthworms (120 for each species) were collected two weeks prior to the beginning of the experiment.

### Food quality

The effects of foof quality were assessed by using two different types of plant litter of contrasting nutritional quality: *M. pruriens* (52.4% C, 2.25% N, 23.3 C/N, and 9.67% ash) and maize stubble (52% C, 0.84% N, 61.9% C/N, and 10.3% ash). Both materials were obtained from the MM field, oven-dried at 60˚C for 48 h, and sieved (1 mm).

### Experiment

Growth, sexual maturity, reproduction (cocoons and juveniles), and mortality of *B. pearsei* and *P. corethrurus* were investigated during 100 days using a factorial design with three factors: origin of earthworms (OE), soil moisture (SM), and food quality (FQ). SM involved 3 levels, corresponding to the permanent wilt point (25%), field capacity (42%), and an intermediate level (33%). FQ included three levels: 300 g soil only (S), 294 g soil + 6 g maize stubble (MS), and 294 g soil + 6g *M. pruriens* (MP); the amounts added correspond to those commonly found in both maize monocultures and cultures rotating maize and *M. pruriens*. The earthworm species used belong to two different classes based on origin: Native (*B. pearsei*) and Invasive (*P. corethrurus*).

**Table 1. *F*-values and significance levels (ANOVA) of the interaction of three factors on growth and reproduction of the tropical endogeic earthworm *Pontoscolex corethrurus* and *Balanteodrilus pearsei* at 100 days of culture in soil with low anthropic impact.**

| Independent variable | Biomass adult | | Sexual maturity | | Cocoon | | | | Juveniles | | | |
|---|---|---|---|---|---|---|---|---|---|---|---|---|
| | | | | | Number | | Biomass | | Number | | Biomass | |
| | F | P | F | P | F | P | F | P | F | P | F | P |
| **Origen Earthworm (OE)** | 29.7 | 0.0000 | 22.3 | 0.0000 | 4.4 | 0.0390 | 2238.8 | 0.0000 | 25.1 | 0.0000 | 836.9 | 0.0000 |
| **Soil Moisture (SM)** | 20.4 | 0.0000 | 9.2 | 0.0002 | 22.2 | 0.0000 | 31.7 | 0.0000 | 14.8 | 0.0000 | 35.2 | 0.0000 |
| **Food quality (FQ)** | 191.7 | 0.0000 | 299.4 | 0.0000 | 109.0 | 0.0000 | 12.5 | 0.0004 | 67.0 | 0.0000 | 11.6 | 0.0006 |
| **OE×SM** | 0.03 | 0.9739 | 4.2 | 0.0164 | 4.4 | 0.0161 | 9.3 | 0.0000 | 5.0 | 0.0095 | 13.5 | 0.0000 |
| **OE×FQ** | 5.2 | 0.0068 | 3.2 | 0.0438 | 11.6 | 0.0000 | 3.5 | 0.0609 | 25.2 | 0.0000 | 10.8 | 0.0010 |
| **FQ×SM** | 7.3 | 0.0000 | 2.1 | 0.0897 | 13.4 | 0.0000 | 5.9 | 0.0029 | 11.4 | 0.0000 | 21.7 | 0.0000 |
| **OE×FQ×SM** | 1.1 | 0.3626 | 2.5 | 0.0442 | 6.4 | 0.0002 | 2.5 | 0.0859 | 6.2 | 0.0002 | 15.3 | 0.0000 |

The combination of the three factors and three levels produced nine treatments with five replicates per treatment. Each replicate consisted of a plastic container (12×12×8 cm) containing 300 g dried soil of the corresponding food-soil mixture and soil moisture; two individuals of *B. pearsei* and two of *P. corethrurus* were transferred to each container (Table 1).

Earthworms were washed, dried on paper towels, weighed, and assigned randomly to each treatment. The baseline weight of the 45 replicates from the nine treatments was statistically similar in *B. pearsei* and *P. corethrurus* (76.06 ± 26.1 mg, n = 90 and 66.04 ± 31.1 mg, n = 90, respectively). Containers were incubated at 26 ± 1 °C. Body weight, mortality, clitellum appearance (sexual maturity), and number and biomass of cocoons and juveniles of *B. pearsei* and *P. corethrurus* were recorded at 10-day intervals, and soil was replaced. Before use, fresh soil (including the corresponding food-soil mixture and moisture) was preincubated for 8 days at 26°C in order to trigger litter substrate decomposition. Each cocoon produced was incubated in a petri dish at 26°C; incubation time as well as number and weight of all juveniles hatched were recorded.

## Statistical analysis

Cocoon and juvenile weight, and growth were evaluated through Analysis of Varienace (ANOVA). Mortality, sexual maturity, number of cocoons, and number of juveniles were analyzed using generalized linear models, specifically the Poisson distribution wich is widely used for modelling count data. Differences between means were evaluated with Tukey's HSD. All statistical analyses were perfomed using the Statistica software.

## Results

At 100 days of culture, significant effects were observed between the origin of earthworms (OE), food quality (FQ) and soil moisture (SM), and the interaction between these three factors on sexual maturity, number of cocoons, and number and biomass of juveniles (Table 1).

## Mortality

At the end of the culture, the invasive earthworm (*P. corethrurus*) had a 21.1% mortality rate in the soil treatment (33% and 25% SM), while that of the native earthworm (*B. pearsei*) had only a 1.1% mortality rate in the soil treatment (only 42% SM). In the *M. pruriens* and maize stubble treatments (25%, 33% and 42% SM) no mortality was observed in both earthworm species.

## Growth

Growth of the endogeic earthworms clearly varied in response to EO, FQ, SM, and the EO⊆CF and SM⊆CF interactions (Table 1). At 100 days of culture, the growth of the invasive and native species (*P. corethrurus* and *B. pearsei*, respectively) was higher when food quality increased (Fig 1). In the three FQ levels (soil, maize stubble, and *M. pruriens*) the exotic species showed a faster growth (1.6, 9.4, and 12.3 mg/day, respectively) relative to the native species (0.34, 4.8, and 10.4 mg/day).

## Reproduction

**Sexual maturity (clitellum).** When fed *M. pruriens*, the onset of sexual maturity in *P. corethrurus* and *B. pearsei* occurred at 30 days; when fed maize stubble, sexual maturity was obseved at 30 and 70 days in *P. corethrurus* and *B. pearsei*, respectively.

At 100 days of culture, OE, FQ, SM, and the OE ⊆ CF ⊆ SM interaction significantly affected clitellum development (Table 1). The invasive and native earthworms reached sexual maturity in the treatments with *M. pruriens* (100% and 86.6%) and maize stubble (96.7% and 70.0%), respectively (Fig 2). No individuals reached sexual maturity after 100 days in the soil treatments; however, in the soil treatment with 33% SM, one earthworm of *P. corethrurus* (6.7%) reached sexual maturity at 80 days.

**Cocoon production.** *B. pearsei* and *P. corethrurus* displayed biparental and uniparental sexual reproduction, respectively. On *M. pruriens* treatments (25%, 33% and 42% SM), cocoon production started when *B. pearsei* and *P. corethrurus* reached a mean biomass of 773.5 ± 146.8 mg and 644.7 ± 71.1 mg (average of 25%, 33% and 42% SM), respectively. On maize stubble treatments, it started when *B. pearsei* and *P. corethrurus* reached a mean body weight of 593.0 ± 80.9 mg and 598.5 ± 95.2 mg (average of 25%, 33% and 42% SM), respectively. Cocoon production in *P. corethrurus* was observed in soil (6.7%), maize stubble (53.3%), and *M. pruriens* (86.7%) treatments, but in *B. pearsei* it was observed only in maize stubble (33.3%) and *M. pruriens* (86.7%) treatments.

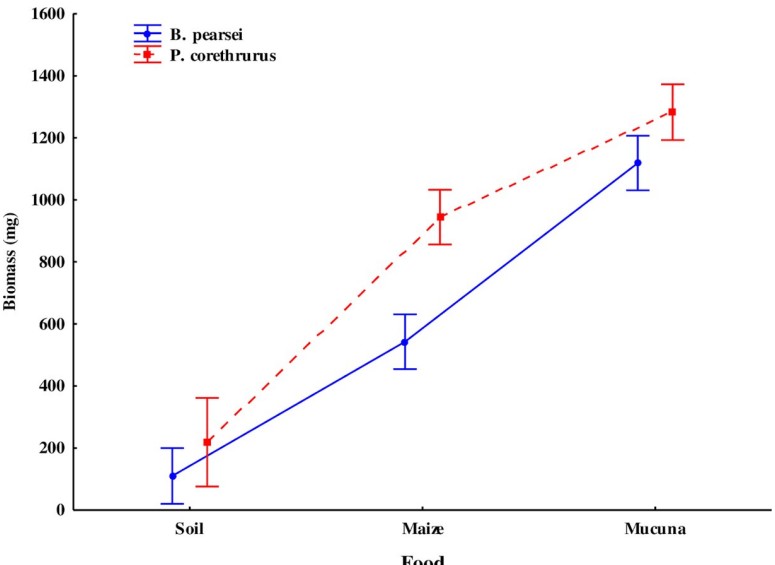

**Fig 1. Biomass of the tropical endogeic earthworms *Pontoscolex corethrurus* (invasive) and *Balanteodrilus pearsei* (native) at 100 days of culture using three diets of different nutritional quality in soil with low anthropic impact.** Vertical lines represent standard error.

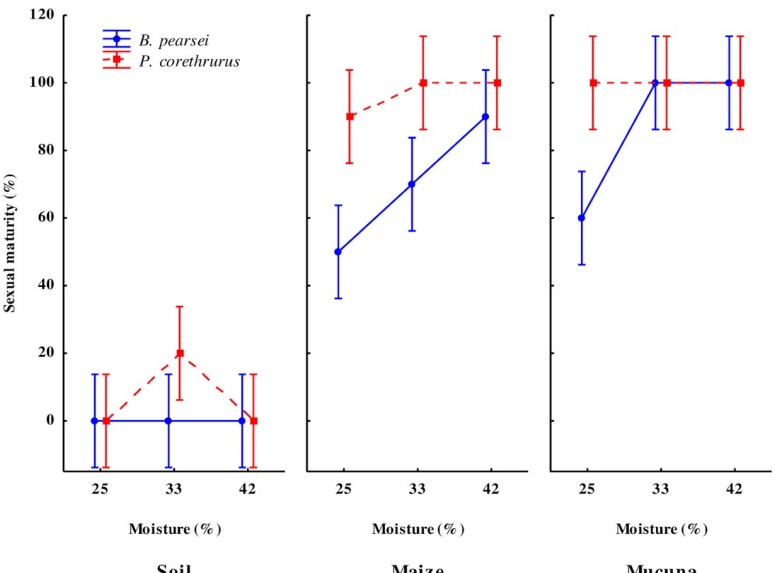

**Fig 2. Sexual maturity (formation of the clitellum) in the tropical endogeic earthworms *Pontoscolex corethrurus* (invasive) and *Balanteodrilus pearsei* (native) at 100 days of culture under the interaction of three diets of different nutritional quality and three moisture content levels in soil with low anthropic impact.** Vertical lines represent standard error.

Mean cocoon production was significantly influenced by EO, CF, SM, and the interaction between these three factors (Table 1). After 100 days of culture, peak mean cocoon production in *B. pearsei* and *P. corethrurus* was observed in *M. pruriens* treatments, with 59.7 ± 40.8 and 35.5 ± 21.5 cocoons (average of 25%, 33% and 42% SM treatments), respectively (Fig 3). When fed maize stubble, *B. pearsei* and *P. corethrhrus* produced 7.9 ± 3.2 and 14.4 ± 9.2 cocoons

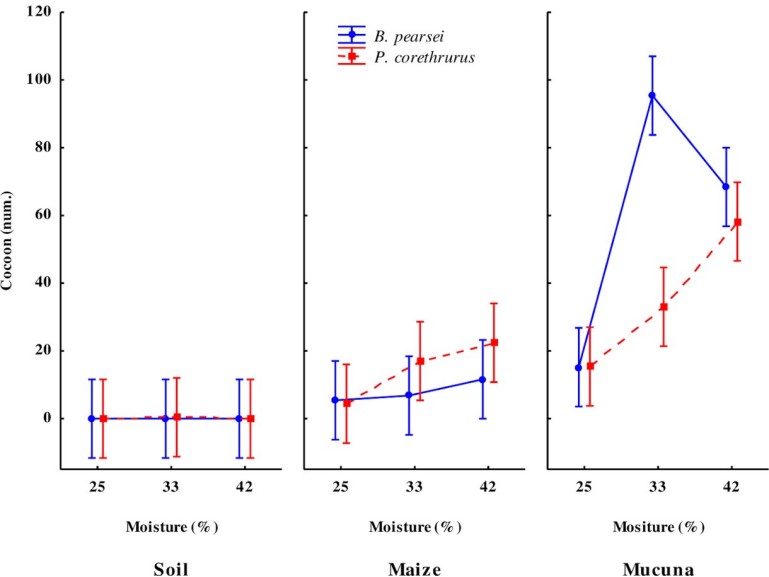

**Fig 3. Number of cocoons produced by the tropical endogeic earthworms *Pontoscolex corethrurus* (invasive) and *Balanteodrilus pearsei* (native) at 100 days of culture under the interaction of three diets of different nutritional quality and three moisture content levels in soil with low anthropic impact.** Vertical lines represent standard error.

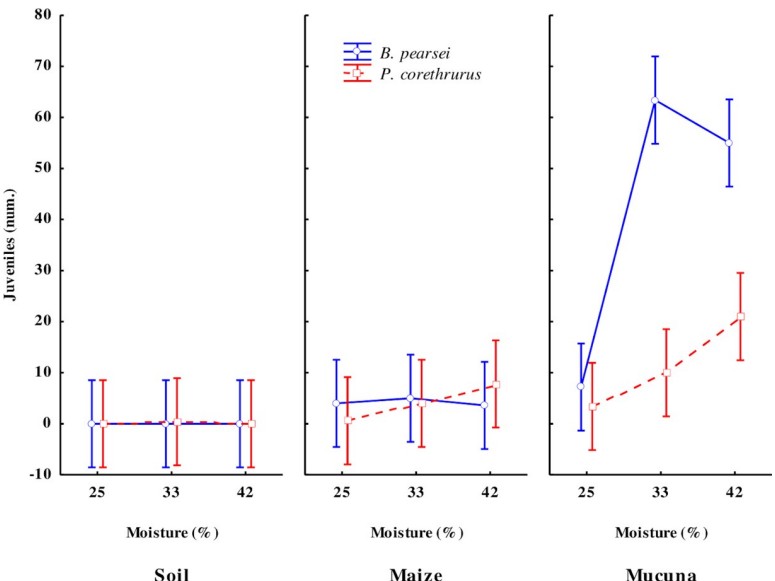

**Fig 4. Number of juveniles hatched from cocoons produced by the tropical endogeic earthworms *Pontoscolex corethrurus* (invasive) and *Balanteodrilus pearsei* (native) at 100 days of culture under the interaction of three diets of different nutritional quality and three moisture content levels in soil with low anthropic impact.** Vertical lines represent standard error.

(average of 25%, 33% and 42% SM treatments), respectively. Finally, when fed soil only (33% SM), *P. corethrurus* (448 mg body weight) produced only two cocoons.

Cocoon biomass varied significantly in response to EO, FQ, SM and the OE x SM and FQ x SM interactions (Table 1). Average cocoon biomass produced by *B. pearsei* and *P. corethrurus* with SM treatments (25%, 33% and 42%) was 10.2 ± 1.4 mg and 27.7 ± 3.7 mg, respectively.

**Juvenile production.** The mean cocoon incubation time was similar among treatments (P > 0.05). In general, mean cocoon incubation time was 20.4 ± 5.2 days (*B. perasei*) and 30.3 ± 2.2 days (*P. corethrhrus*), with one individual hatching per cocoon in all cases. Of the total number of cocoons produced by *B. pearsei* and *P. corethrurus* in *M. pruriens* and corn stubble treatments, the average number of hatched juveniles was 64.7 ± 16.6% and 29.5 ± 7.0% (average of 25%, 33% and 42% SM treatments) and 59.5 ± 24.7 and 24.0 ± 10.6 (average of 25%, 33% and 42% SM treatments), respectively.

The number of hatched juveniles of *B. pearsei* and *P. corethrurus* varied significantly with OE, CF, SM, and the interaction between these three factors (Table 1; Fig 4). The mean number of hatched juveniles of *B. pearsei* and *P. corethrurus* increased in adults fed *M. pruriens*, as well as with increasing soil moisture (mean 59.7±40.8 and 35.5±21.5 individuals, respectively), and corn stubble (mean 7.9 ± 3.3 and 14.6 ± 9.2 individuals).

At hatching, in the *M. pruriens* and corn stubble treatments, mean biomass of *P. corethrurus* juveniles (21.2 ± 1.0 and 18.6 ± 7.4 mg, respectively) was higher vs. *B. pearsei* juveniles (8.5 ± 0.7 and 8.5 ± 1.3 mg, respectively).

## Discussion

Domesticated, wild populations respond to changing selective pressures, which are reflected in their adaptation to agricultural niches [2, 46]. From an ecological perspective, the endogeic earthworm *P. corethrurus* resembles non-domesticated species or weeds given its strong profile (invading species) regarding growth rate, fertility, plasticity, interspecific competition, and

environmental tolerance [7, 8, 9, 26, ]. This suggests that the four *P. corethrurus* ecotypes described by Taheri et al. [47] are likely the result of the selective forces imposed by cultivation, agricultural practices, and industrial and urban activities [20]. In the present study, soil in the habitat for *B. pearsei* was observed to restrain the presence of *P. corethrurus*.

The conversion of the Amazon forest to pastures led to the homogenization of soil biota [3, 48]. The potential resistance of soil (i.e., predators, low species richness, etc.) to earthworms has been documented [15, 49, 50]. For instance, the endogeic tropical earthworm *Millsonia anomala* from the savannah was unable to prosper in forest soil [49], similar findings have been reported with *P. corethrurus* from fallow (slash-and-burn) to mature forest [35]. Also, the shift in vegetation from grass to woody plants decreaced in the density and biomass of *P. corethrurus* [51]. Our results showed that the survival of *P. corethrurus* was lower in the environment where *B. pearsei* thrives, maybe due to a negative interaction with a more diverse edaphic microbiome [49, 50, 52], because it has been suggested that *P. corethrhrus* has a high ability to utilize soil organic resources as an energy source [39].

Earthworms harbor symbiotic microbiomes that are essential for their life history in the nephridia (excretory organs), and cocoons in tropical species such as *P. corethrurus* is poorly studied [53–58]. The microbiome is known to improve the nutritional status of low-quality diets [57–58]. For example, Topoliantz and Ponge [35] observed that the behaviour of two populations of *P. corethrurus* separated along the Maroni river (French Guiana, South America) differed significantly: fallow populations produced more cast on charcoal in the presence of forest soil, while the casting activity of the forest population was higher on soil regardless of the soil origin. Our findings show that *P. corethrurus* and *B. pearsei* differ in their diet preference (*M. pruriens*, corn stubble, and control), i.e., the invasive species displayed faster growth than the native species when nutritional quality improved. This suggests that *P. corethrurus* consumes and degrades a greater variety of organic materials given its greater ability (efficiency), evidenced by: a) producing endogenous cellulases [59–62]; b) its association with the gut microbiota [63–66]; c) gene expression (transcriptome) that contribute to the adaptation of its digestive system [65]; d) improving its digestion efficiency according to the type of cecum [59, 67]; and e) its association with nephridial bacteria [50, 68, 69].

It is known that in diets of low nutritional quality, mutualistic bacteria residing in earthworm nephridia (in 19 of 23 species studied) provide vitamins to its host, stimulate earlier sexual maturity, and contribute to pesticide detoxification [56, 57, 58, 60, 70, 71]. The results reported here showed that the invasive species of smaller size (biomass) fed on a lower nutritional diet (*M. pruriens* > corn stubble > soil) reached sexual maturity earlier than the native earthworm. This suggests that the nephridial symbionts of *P. corethrurus* are generalists, while those of *B. pearsei* are specialists.

Earthworms produce external cocoons that are colonized by bacteria from parents and soil [vertical and horizontal transmission, respectively 53, 58] and coul be used as biovectors for the introduction of benefical bateria [55]. In a new habitat, cocoons of invasive earthworms may be affected by the native microbiota, but they can survive if they carry a parental microbial inoculum. Our results show that *P. corethrurus* produced cocoons when fed either of the three diets, while *B. pearsei* fed the control diet (only soil) failed to produce cocoons. In contrast, cocoons of *P. corethrurus* had a low hatching rate (births), which was lower (diet with *M. pruriens*) compared to *B. pearsei*. These results suggest the absence and/or loss of parental symbionts bacteria, i.e., the loss of a parental care strategy to control predators, detoxify nitrogenous wastes, conserve nitrogen, and supply vitamins and essential cofactors to the offspring [55, 56, 57, 68, 69, 70, 72]. Thus, the likely symbiotic evolution of *P. corethrurus* with the microbiome (gut, nephridia and cocoons) should be explored as a source of biogeography and phylogenetic

information [11, 57, 68, 70, 71, 73, 74]. That is, we could ". . .explain why *P. corethrurus* is rare or absent in undisturbed lands" [39].

The human-mediated translocation of species dates back to the Late Pleistocene [2, 5, 75]. Invasive plant species are usually divided in two groups according their residence time: archaeophytes were found from 1500 AD, and neophytes are found after this date [76]. This approach can contribute to elucidate the history of the invasion of *P. corethrurus* in Mexico. Until now, only two ecotypes have been recorded [47] and the criptic linage used in this study corresponds to L1 (the most widespread). The origin of *P. corethrurus* may be related to anthropogenic soil formation ("terras mulatas" and "terras pretas"). The domestication of manioc (bitter and sweet) and peach palm staple food that facilitated sedentary lifestyles in the Amazon region [5, 27, 28, 29, 32] has evolved to the point that we cannot recognize the predecessors of *P. corethrurus*, as evidenced by the recent designation of the *P. corethrurus* neotype from an anthropogenic environment [18] and temperate climate [77], and by the ambiguity used for assigning its place of origin [12, 78].

Based on the results reported here, we conclude that the invasive tropical earthworm *P. corethrurus* had lower survival and cocoons hatching rates (offspring) in the agro-ecological niche of the native endogeic earthworm, i.e., a finding consistent with the absence of *P. corethrurus* in parcels where maize- and *M. pruriens* crop rotation is practiced, as well as in pastures and other traditional tropical agroecosystems [40, 41, 42, 43, 44, 45]. This suggests that *P. corethrurus* is an invasive species that thrives far from its natural status, i.e., has no wild ancestry in the study area. Therefore, it is important to determine the preference of the four *P. corethrurus* ecotypes [47] in terms of soil type, cultivation, response to stressors and climate change.

## Supporting information

**S1 Table. Results fitting linear model of the earthworm biomass.**
(PDF)

**S2 Table. Results fitting logistic model of sexual maturity of the earthworms.**
(PDF)

**S3 Table. Results fitting zero inflated poisson of the earthworms cocoons.**
(PDF)

## Acknowledgments

The authors thank Mario M. Osorio-Arce, Angel Ramos-Sánchez and Efraín Hernández-Xolocotzi, promoters of agroecology in Mexico. In addition, we are grateful to anomymous reviewers and Diana Pérez-Staples for valuable comments and careful revision of the manuscript.

## Author Contributions

**Conceptualization:** Angel I. Ortíz-Ceballos.

**Formal analysis:** Angel I. Ortíz-Ceballos, Diana Ortiz-Gamino, Antonio Andrade-Torres, Paulino Pérez-Rodríguez, Maurilio López-Ortega.

**Investigation:** Angel I. Ortíz-Ceballos.

**Methodology:** Angel I. Ortíz-Ceballos.

**Project administration:** Angel I. Ortíz-Ceballos.

**Resources:** Angel I. Ortíz-Ceballos.

**Writing – original draft:** Angel I. Ortíz-Ceballos, Diana Ortiz-Gamino, Antonio Andrade-Torres, Paulino Pérez-Rodríguez, Maurilio López-Ortega.

**Writing – review & editing:** Angel I. Ortíz-Ceballos.

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
