## [Decision Letter · Decision Letter 0]

5 Jul 2019

PONE-D-19-15357

Pontoscolex corethrurus: a Homeless Invasive Tropical Earthworm?

PLOS ONE

Dear Ceballos

Thank you for submitting your manuscript to PLOS ONE. After careful consideration, we feel that it has merit but does not fully meet PLOS ONE’s publication criteria as it currently stands. Therefore, we invite you to submit a revised version of the manuscript that addresses the points raised during the review process.

1]The authors have done an interesting study determining the reason as to why P. corethrurus is unable to invade the agricultural niche of B. pearsei?

2]The design of experiment is very appropriate and accurate.

3]But there are some loop holes that need to be plugged for eg: the figure numbers need to be checked and rectified

4]English language needs to be checked by someone with good command over language for clarification in the manuscript

5]Diiscussion should be written more precisely and directly analysing the results obtained in the experiments performed.

6]Other comments have been high lighted in the manuscript and also by the reviewers ,

7]All the comments must be answered to and necessary rectifications made in the manuscript based on the reviewers comments

Therefore manuscript can be accepted for publication only after the major revision.

We would appreciate receiving your revised manuscript by 25 July 2019. To enhance the reproducibility of your results, we recommend that if applicable you deposit your laboratory protocols in protocols.io, where a protocol can be assigned its own identifier (DOI) such that it can be cited independently in the future. For instructions see: http://journals.plos.org/plosone/s/submission-guidelines#loc-laboratory-protocols

We look forward to receiving your revised manuscript.

Kind regards,

Tunira Bhadauria, Ph.D.

Academic Editor

PLOS ONE

Journal Requirements:

2. Please upload a copy of Figures 5 and 6, to which you refer in your text on page 11. If the figure is no longer to be included as part of the submission please remove all reference to it within the text.

3. In your Methods section, please provide additional location information of the collection sites, including geographic coordinates for the data set if available.

4. In your Methods section, please provide additional information regarding the permits you obtained for the work. Please ensure you have included the full name of the authority that approved the field site access and, if no permits were required, a brief statement explaining why.

5. Please include your tables as part of your main manuscript and remove the individual files. Please note that supplementary tables (should remain/ be uploaded) as separate "supporting information" files

Additional Editor Comments:

1]The authors have done an interesting study determining the reason as to why P. corethrurus is unable to invade the agricultural niche of B. pearsei?

2]The design of experiment is very appropriate and accurate.

3]But there are some loop holes that need to be plugged for eg: the figure numbers need to be checked and rectified

4]English language needs to be checked by someone with good command over language for clarification in the manuscript

5]Diiscussion should be written more precisely and directly analysing the results obtained in the experiments performed.

6]Other comments have been high lighted in the manuscript and also by the reviewers ,

7]All the comments must be answered to and necessary rectifications made in the manuscript based on the reviewers comments

Therefore manuscript can be accepted for publication only after the major revesion.

Reviewers' comments:

Reviewer's Responses to Questions

**Comments to the Author**

1. Is the manuscript technically sound, and do the data support the conclusions?

Reviewer #1: Partly

2. Has the statistical analysis been performed appropriately and rigorously? 

Reviewer #1: Yes

3. Have the authors made all data underlying the findings in their manuscript fully available?

Reviewer #1: Yes

4. Is the manuscript presented in an intelligible fashion and written in standard English?

Reviewer #1: Yes

5. Review Comments to the Author

Reviewer #1: The study asks an interesting question given at the end of the introduction—why hasn’t P. corethrurus invaded the agricultural niche of B. pearsei? The experiment seems well designed. However, there are some items in the test that need to be corrected (e.g. figure numbers), some revisions for clarity are needed, and the Discussion needs revisions to make it more directly relevant to the experiment that was conducted. These are explained below.

Page 4 line 3, 100-120 of the species?

Last sentence page 4, confusing—are you switching back to P. corethrurus here?

Page 6, second to last line, 61.9 C/N does not have a percent—it’s a ratio

Page 8, mortality, please state the mortality results more systematically, instead of a few selected treatments.

Page 8-9. Figure 1 does not show growth rates, it shows biomass at 100 days with three food sources. In the second sentence it should say ‘At 100 days the biomass of invasive and native species…’ The last sentence appears to actually present growth rate data. Biomass at 100 days is related to the growth rate, but is not the same thing.

Middle of page 10. Fig. 4 does not show cocoon biomass.

Page 10-11, there are no Figs 5 and 6 in the manuscript that I have.

Discussion:

First sentence is incomplete; either say what happens when populations respond to domestication, or delete ‘When’ at the beginning of the sentence.

Page 13-14, 275 and 176 species of bacteria?

Page 15, Levis et al 2018 (not 208).

Most of the Discussion (although very interesting) is not about what was studied, and connections between the experiment presented in the results section and discussion topics are weak. For example, does your P. corethrurus belong to one of the four agrotypes mentioned on page 16? Do you know which bacterial taxonomic groups are present in your P. corethrurus?

The statements in the concluding paragraph on page 17 have no direct evidence from your study. Again, it is interesting, and well written, but very speculative as it relates to the study.

There is almost a review paper about evolutionary/agro-history of the earthworm species embedded in the Discussion. Perhaps the authors should consider expanding this and publishing it separately as a review paper! However, for purposes of the manuscript under review, the connections to the experiment are too weak, and this review material should be greatly shortened to one or two paragraphs. In place of most of this material there should be more discussion points related directly to the experimental evidence—for example comparison to survival rates, attainment of sexual maturity, biomass and cocoon production from other studies of these two earthworm species. Also, the conclusion should have a succinct summary about how the evidence from the experiment answers the question posed at the end of the introduction.

6. PLOS authors have the option to publish the peer review history of their article (what does this mean?). If published, this will include your full peer review and any attached files.

Reviewer #1: No

---

## [Author Response · Author response to Decision Letter 0]

26 Jul 2019

Rebuttal Letter

General remarks

All the corrections indicated referees (Reviewer # 1 and Editorial Board Editor) was incorporated on the manuscript. In green, sentences and words eliminated; in red, sentences, words incorporated, and only in the text of the manuscript. 

Specific points:

Introduction

Comments:

Page 4 line 3, 100-120 of the species? 

Last sentence page 4, confusing—are you switching back to P. corethrurus here?  Page Line

Response: 

We agree with these comments. We include and delete the following information suggested:

Include:

It has been documented that 3 % of the diversity of earthworms are invasive species (Dupont et al. 2012).

Delete:

It has been documented that 3 % (100-120) of the diversity of earthworms are invasive species (Hendrix 2008), and stating that just a few of them had a negative impact on terrestrial agroecosystems would not be an exaggeration (Simberloff 2009; Hendrix et al. 2006) 

3 

70-71

Results

Comments:

Page 8, mortality, please state the mortality results more systemaWcally, instead of a few selected treatments.  Page Line

Response

We agree with these comments. We include and delete the following information suggested:

Include:

At 100 days of culture, significant effects were observed between the origin of earthworms (OE), food quality (FQ) and soil moisture (SM), and the interaction between these three factors on sexual maturity, number of cocoons, and number and biomass of juveniles (Table 1). 

Mortality

At the end of the culture, the invasive earthworm (P. corethrurus) had a 21.1 % mortality rate in the soil treatment (33 % and 25 % SM), while that of the native earthworm (B. pearsei) had only a 1.1 % mortality rate in the soil treatment (only 42 % SM). In the M. pruriens and maize stubble treatments (25 %, 33 % and 42 % SM) no mortality was observed in both earthworm species.

Delete:

At 100 days of culture, significant effects were observed between the origin of earthworms (OE), food quality (FQ), soil moisture (SM) and mortality, and the interaction between these three factors on sexual maturity, number of cocoons, and number and biomass of juveniles. The invasive earthworm (P. corethrurus) had a 21.1 % mortality rate in the soil treatment (33 % and 25 % SM). In contrast, the native earthworm (B. pearsei) had only a 1.1 % mortality rate in M. pruriens (only 42 % SM). 

8 

175-183

Discussion

Comments:

Discussion should be wrigen more precisely and directly analysing the results obtained in the experiments performed.  Page Line

Response:

We agree with these comments. We delete the following information suggested:

Included:

This suggets that the four P. corethrurus ecotypes described by Taheri et al. (2018a) are likely the result of the selective forces imposed by cultivation, agricultural practices, and industrial and urban activities (Taheri et al. 2018b).

Eliminated:

a finding consistent with the absence of P. corethrurus in parcels where maize- and M. pruriens crop rotation is practiced, as well as in pastures and other traditional tropical agroecosystems (Lavelle et al. 1981; Ortiz- Ceballos et al. 2004; Huerta et al. 2006; Marichal et al. 2010; Fragoso et al. 2016; Ortiz-Gamino et al. 2016). 

Included:

, similar findings have been reported with P. corethrurus from fallow (slash-and-burn) to mature forest (Topoliantz and Ponge 2005).

, because it has been suggested that P. corethrhrus has a high ability to utilize soil organic resources as an energy source (Lavelle et al. 1987). 

Delete:

Also, in a Glenrock soil (southern Australia), the survival of the European endogeic earthworm Aporrectodea trapezoides was found in association with soil bacteria (Davidson et al 2013; Menezes et al. 2018).

Include:

For example, Topoliantz and Ponge (2005) observed that the behaviour of two populations of P. corethrurus separated along the Maroni river (French Guiana, South America) differed significantly: fallow populations produced more cast on charcoal in the presence of forest soil, while the casting activity of the forest population was higher on soil regardless of the soil origin.

Delete:

Symbiosis, defined as the interaction between two different organisms living in close physical association, has been acknowledged as a source of evolutionary innovation that allows hosts to exploit otherwise inaccessible niches (Laland et al. 1999; Lund et al. 2010b; Aira et al. 2018). The hologenome (sum of the genetic information of the host and its microorganisms) theory of evolution is based on four generalization: a) all animals and plants establish symbiosis with microorganims; b) microorganisms can be transmitted between generations with fidelity; c) symbiosis affects the fitness of holobionts in their environment; d) genetic variation in holobionts can be enhanced by incorporating different symbiont populations and can change rapidly under enviromental stress (Zilber-Rosenberg and Rosenberg 2008). 

Include:

Thus, the likely symbiotic evolution of P. corethrurus with the microbiome (gut, nephridia and cocoons) should be explored as a source of biogeography and phylogenetic information (Lund et al. 2010a; Brussaard et al. 2012; Davidson et al. 2013; Møller et al. 2015; Zwarycz et al. 2015; Schult et al. 2016; Marchán et al. 2018). That is, we could “…explain why P. corethrurus is rare or absent in undisturbed lands” (Lavelle et al. 1987).

Delete:

In Eisenia andrei and E. fetida, 275 and 176 bacteria were observed in their cocoons, respectively (Menezes et al. 2018); however, these were dominated by three vertically transmitted (parental) symbionts: Microbacteriaceae, Verminephrobacter, and Ca. Nephrothrix. For example, in a sterile environment, cocoons of M. anomala failed to develop, suggesting a functional relationship between nephridial bacteria of earthworms and soil microbiota (Gilot-Villenave 1994). Besides, it has been noted that during embryogenesis, the nephridial symbiont Verminephrobacter (Betaprototeobacteria) contributes to the biosynthesis of vitamins and the elimination of nitrogen wastes from cocoons, and conserving nitrogen (Schramm et al. 2003; Davidson and Stahl 2006; Lund et al. 2010b; Davidson et al. 2013; Møller et al. 2015). 

Include:

Until now, only two ecotypes have been recorded (Taheri et al. 2018a) and the criptic linage used in this study corresponds to L1 (the most widespread). The origin of P. corethrurus may be related to anthropogenic soil formation (“terras mulatas” and “terras pretas”). The domestication of manioc (bitter and sweet) and peach palm staple food that facilitated sedentary lifestyles in the Amazon region (Lodge 1993; Glaser et al. 2000; Arroyo-Kalin 2010; Clement et al. 2015; Watling et al. 2018; Levis et al. 2018) has evolved to the point that we cannot recognize the predecessors of P. corethrurus, as evidenced by the recent designation of the P. corethrurus neotype from an anthropogenic environment (James et al. 2019) and temperate climate (Peel 2007), and by the ambiguity used for assigning its place of origin (Righi 1984; Dupont et al. 2012). 

Delete:

Pyšek et al (2005) observed that archeophyte weeds are common in crops introduced at the beginning of agriculture (cereals), but are poorly represented in crops introduced relatively recently (rape, maize), where neophyte weeds are most numerous. 

It is documented that an exchange of domesticated plants (manioc, maize, cacao and others), as well as recently introduced plants (livestock, pastures, coffee, sugar cane and others), took place between the Amazon area and Mesoamerica through agricultural packages that may have also included P. corethrurus as well as weeds, mice, insects, etc. (Denevan 1992; Boivin et al 2016; Levis et al. 208). 

Delete:

Therefore, the four P. corethrurus ecotypes described by Taheri et al. (2018b) are likely the result of the selective forces imposed by cultivation, agricultural practices, and industrial and urban activities. That is, this species has evolved to develop resistance to herbicides, pesticides, heavy metals, and hydrocarbons, among others (Taheri et al. 2018b). The above may explain why the paradigm of Beddard (1900) " ...tropical earthworms invade only tropical regions (Neotropical) ..." is no longer valid, as P. corethrurus has been recorded across the five biogeographic regions: Nearctic (Gates 1954, Gates 1972, Blakemore 2009), Palearctic (Omodeo et al. 2003; Blakemore et al. 2006; Reynolds and Jones 2006, Sherlock and Carpenter 2009, Blakemore 2009, Cunha et al. 2014, Rota and Jong 2015; Taheri et al. 2018), Ethiopian (Plisko 2001), Indian (Tsai et al. 2000; Blakemore 2009; Singh et al. 2018; Subedi et al. 2018), and Australian (Blakemore 2009). The tropical microclimatic model of Lavelle et al. (1987) for the growth and development of P. corethrurus (20-30 °C) is also not applicable to all four P. corethrurus agrotypes (James et al. 2019), since a revision of databases and the literature revealed that this species has been collected in Neotropical latitudes with temperate climate (10- 20 °C; 1200 to 2000 m a.s.l.; Köppen 2011; Peel 2007; Volken and Brönnimann 2011) from México (Fragoso 2018; Juárez-Ramón and Fragoso 2014, Ortiz-Gamino et al. 2016), Lesser Antilles (Hendrix et al. 2006), Colombia (Gutiérrez-Sarmiento and Cardona 2014), Brasil (Müller 1857; Bunch et al. 2011; Ferreira et al. 2018; Steffen et al. 2018), Argentina (Mischis and Righi 1999; Mischis 2007), Madagascar (Chapuis-Lardy et al. 2010; Villenave et al. 2010), South África (Plisko 2001, Janion-Scheepers et al. 2016) e India (Singh et al. 2018; Subedi et al. 2018). 

The origin of P. corethrurus may be related to anthropogenic soil formation (“terras mulatas” and “terras pretas”) and the domestication of manioc (bitter and sweet) and peach palm staple food that facilitated sedentary lifestyles in the Amazon region (Lodge 1993; Glaser et al. 2000; Arroyo-Kalin 2010; Clement et al. 2015; Watling et al. 2018; Levis et al. 2018), and has evolved to the point that we cannot recognize their wild predecessors, as evidenced by the recent designation of the P. corethrurus neotype from an anthropogenic environment (Müller 1857; James et al. 2019), and by the ambiguity used for assigning its place of origin (Righi 1984; Dupont et al. 2012). 

Cryptic invasions occur frequently, often go unnoticed, and are hard to recognize. Also, it is known that cryptic species may display markedly different responses to a given stimulus or stressor (Liebeke et al. 2014; Schult et al. 2016; Morais and Reichard 2018). Therefore, it is important to classify the four P. corethrurus ecotypes (Taheri et al. 2018a; James et al. 2019) as separate evolutionary entities over the residence time, and determine the preference of each ecotype in terms of soil type, culture, and climate, and response to stimulus or stressors (pesticides, herbicides, heavy metals, etc.), among others. To this end, transcriptomes (Rad-Seq) may be used to elucidate the gene flow across cryptic lineages (Puga-Freitas et al. 2015; Schult et al. 2016; Morais and Reichard 2018). In addition, epigenetics could be used to investigate the functional adaptations of P. corethrurus lineages (Kille et al. 2013; Liu et al. 2017; Ponesakki et al. 2017; Fernández-Marchán et al. 2018). Last, potential symbionts may be found for each agrotype, including bacteria, nematodes, enchytraeids and other life forms associated with earthworms (Coates 1990; Fernández-Marchán et al. 2018). 

Comment: 

the conclusion should have a succinct summary about how the evidence from the experiment answers the question posed at the end of the introduction. 

Response:

We agree with these comments. We delete the following information suggested

Include:

Based on the results reported here, we conclude that the lower survival and hatching rates of P. corethrurus cocoons (offspring) in the agro-ecological niche of B. pearsei; it is, a finding consistent with the absence of P. corethrurus in parcels where maize- and M. pruriens crop rotation is practiced, as well as in pastures and other traditional tropical agroecosystems (Lavelle et al. 1981; Ortiz-Ceballos et al. 2004; Huerta et al. 2006; Marichal et al. 2010; Fragoso et al. 2016; Ortiz-Gamino et al. 2016). This suggests that P. corethrurus is an invasive species that thrives far from its natural status, i.e., has no wild ancestry in the study area. Therefore, it is important to determine the preference the four P. corethrurus ecotypes (Taheri et al. 2018a) in terms of soil type, cultivation, response to stressors and climate change.

Delete:

Based on the results reported here, we conclude that the lower survival and hatching rates of P. corethrurus cocoons (offspring) are associated with its symbiotic bacteria, coupled with a higher diversity of the edaphic microbiome in the agro-ecological niche of B. pearsei. This suggests that P. corethrurus is an exotic species that thrives far from its natural status, i.e., has no wild ancestry in the study area. For this reason, the likely symbiotic eco-evolution of P. corethrurus with the microbiome in its gut, nephridia and cocoons should be explored as a source of biogeography and phylogenetic information (Lund et al. 2010a; Brussaard et al. 2012; Davidson et al. 2013; Møller et al. 2015; Zwarycz et al. 2015; Schult et al. 2016; Fernández- Marchán et al. 2018), i.e., going back to the roots (Philippot et al. 2013; Pérez-Jaramillo et al. 2016). 

11

11

12

12

13-14

14

14-15

249-251

259-260

265-266

271-275

305-310

315-324

325-333

Figures

Figures in the text:

Comments:

Page 8-9. Figure 1 does not show growth rates, it shows biomass at 100 days with three food sources. In the second sentence it should say ‘At 100 days the biomass of invasive and naWve species...’ 

We agree with these comments. We delete the following information suggested:

Include:

Fig. 1. Biomass of the tropical endogeic earthworms Pontoscolex corethrurus (invasive) and Balanteodrilus pearsei (native) at 100 days of culture using three diets of different nutritional quality in soil with low anthropic impact. Vertical lines represent standard error.

Delete:

Fig. 1. Growth rate of the tropical endogeic earthworms Pontoscolex corethrurus (invasive) and Balanteodrilus pearsei (native) after 100 days of culture using three diets of different nutritional quality in soil with low anthropic impact. Vertical lines represent standard error. 

Comments:

Middle of page 10. Fig. 4 does not show cocoon biomass. Page 10-11, there are no Figs 5 and 6 in the manuscript that I have. 

We agree with these comments. We delete the following information suggested:

Include:

Fig. 1. Biomass of the tropical endogeic earthworms Pontoscolex corethrurus (invasive) and Balanteodrilus pearsei (native) at 100 days of culture using three diets of different nutritional quality in soil with low anthropic impact. Vertical lines represent standard error.

Fig. 2. Sexual maturity (formation of the clitellum) in the tropical endogeic earthworms Pontoscolex corethrurus (invasive) and Balanteodrilus pearsei (native) at 100 days of culture under the interaction of three diets of different nutritional quality and three moisture content levels in soil with low anthropic impact. Vertical lines represent standard error.

Fig. 3. Number of cocoons produced by the tropical endogeic earthworms Pontoscolex corethrurus (invasive) and Balanteodrilus pearsei (native) at 100 days of culture under the interaction of three diets of different nutritional quality and three moisture content levels in soil with low anthropic impact. Vertical lines represent standard error.

Fig. 4. Number of juveniles hatched from cocoons produced by the tropical endogeic earthworms Pontoscolex corethrurus (invasive) and Balanteodrilus pearsei (native) at 100 days of culture under the interaction of three diets of different nutritional quality and three moisture content levels in soil with low anthropic impact. Vertical lines represent standard error.

Delete:

We remove figures 4 (cocoon biomass) and 6 (juveniles biomass) from the text; that is, there were many figures.

References

We included and delete the following references: Page Line

Include:

1. Fragoso, G.C. (2018). Importancia de las lombrices de tierra (Oligochaeta) en el monitoreo de áreas prioritarias de conservación del centro, este y sureste de México. CONABIO. https://doi.org/10.15468/omvnpi accessed via GBIF.org on 2019-05-01

2. Kim, J.S., Sparovek, G., Longo, R.M., de Melo, W.J., & Crowley, D. (2007). Bacterial diversity of terra and pristine forest soil from Western Amazon. Soil Biology & Biochemistry 39: 684-690.

3. Lavelle, P., Maury, M.E., & Serrano, V. (1981). Estudio cuantitativo de la fauna del suelo en la región de Laguna Verde, Veracruz. Publicaciones Instituto de Ecología (México) 6:75-105.

4. Lima H.N., Lima, Schaefer E.R., Mello J.W.V., Gilkes R.J., & Ker J.C. (2002). Pedogenesis and pre-Colombian land use of “Terra Preta Anthrosols” (“Indian black earth”) of Western Amazonia. Geoderma 110: 1-17.

5. Lodge, D.M. (1993). Biological invasions: lessons for ecology. Trends in Ecology & Evolution 8(4): 133-137

6. Ortiz-Ceballos, A.I., Fragoso, C. (2004). Earthworm populations under tropical maize cultivation: the effect of mulching with Velvetbean. Biol. Fert. Soils 39, 438-445.

7. Schaefer, C.E.G.R., Lima, H.N., Gilkes, R.J., & Mello, J.W.V. (2004). Micromorphology and electron microprobe analysis of phosphorus and potassium forms of an Indian Black Earth (IBE) Anthrosol form Western Amazonia. Australian Journal of Soil Research 24(4): 401-409.

8. Topoliantz, S., & Ponge, J.F. (2005). Charcoal consumption and casting activity by Pontoscolex corethrurus (Glossoscolecidae). Applied Soil Ecology 28: 217-224.

Eliminated:

1. Beddard, F.E. (2011). Earthworms and their allies. Earthworms and their allies. https://doi.org/10.5962/bhl.title.17625

2. Chapuis-Lardy, L., Brauman, A., Bernard, L., Pablo, A. L., Toucet, J., Mano, M.J., Weber, L., Brunet, D., Razafimbelo, T., Chotte, J.L., & Blanchart, E. (2010). Effect of the endogeic earthworm Pontoscolex corethrurus on the microbial structure and activity related to CO2 and N2O fluxes from a tropical soil (Madagascar). Applied Soil Ecology, 45(3), 201–208. https://doi.org/10.1016/j.apsoil.2010.04.006

3. Coates, K.A. (1990). Redescriptions of Aspidodrilus and Pelmatodrilus, enchytraeids (Annelida, Oligochaeta) ectocommensal on earthworms. Canadian Journal of Zoology, 68(3), 498–505. https://doi.org/10.1139/z90-073

4. Cunha, L., Montiel, R., Novo, M., Orozco-Terwengel, P., Rodrigues, A., Morgan, A.J., & Kille, P. (2014). Living on a volcano’s edge: Genetic isolation of an extremophile terrestrial metazoan. Heredity, 112(2), 132–142. https://doi.org/10.1038/hdy.2013.84

5. Fragoso, G.C. (2018). Importancia de las lombrices de tierra (Oligochaeta) en el monitoreo de áreas prioritarias de conservación del centro, este y sureste de México. CONABIO. https://doi.org/10.15468/omvnpi accessed via GBIF.org on 2019-05-01

6. Gates, G.E. (1954). Exotic earthworms of the United States. Bulletin of the Museum of Comparitive Zoology at Harvard College, 111(6), 216–258.

7. Gates, G.E. (1972). Burmese earthworms. An introduction to the systematics and biology of Megadrile Oligochaetes with special reference to Southeast Asia. Transactions of the American Philosophical Sciety, 62, 1-326

8. Gutiérrez-Sarmiento, M.C., & Cardona, C.M. (2014). Caracterización ecológica de las lombrices (Pontoscolex corethrurus) como bioindicadoras de suelos compactados bajo condiciones de alta humedad del suelo con diferentes coberturas vegetales (Zipacón, Cundinamarca). Revista Científica, 2(19), 41. https://doi.org/10.14483/23448350.6493

9. Hendrix, P.F. (2006). Biological invasions belowground-earthworms as invasive species. In Biological Invasions Belowground: Earthworms as Invasive Species (pp. 1–4). https://doi.org/10.1007/978-1-4020-5429-7_1

10. Hendrix, P.F., Callaham MacA., Drake, Jr.J.M., Huang, C.-Y., James, S.W., Snyder, B.A., & Zhang, W. (2008). Pandora’s Box Contained Bait: The Global Problem of Introduced Earthworms. Annual Review of Ecology, Evolution, and Systematics, 39(1), 593–613. https://doi.org/10.1146/annurev.ecolsys.39.110707.173426

11. Janion-Scheepers, C., Measey, J., Braschler, B., Chown, S.L., Coetzee, L., Colville J.F., Dames, J., Davies, A.B., Davies, S.J., Davies, A.L.V.,…Wilson J.R.U. (2016). Soil biota in a megadiverse country: Current knowledge and future research directions in South Africa. Pedobiologia, 59(3), 129–174. https://doi.org/10.1016/j.pedobi.2016.03.004

12. Kille, P., Andre, J., Anderson, C., Ang, H.N., Bruford, M.W., Bundy, J.G., … Spurgeon, D. J. (2013). DNA sequence variation and methylation in an arsenic tolerant earthworm population. Soil Biology and Biochemistry 57, 524-532. https://doi.org/10.1016/j.soilbio.2012.10.014

13. Köppen, W., Volken, E., & Brönnimann, S. (2011). The thermal zones of the Earth according to the duration of hot, moderate and cold periods and to the impact of heat on the organic world. Meteorologische Zeitschrift, 20(3), 351–360. https://doi.org/10.1127/0941-2948/2011/105

14. Laland, K.N., Odling-Smee, F.J., & Feldman, M.W. (1999). Evolutionary consequences of niche construction and their implications for ecology. Proceedings of the National Academy of Sciences, 96(18), 10242–10247. https://doi.org/10.1073/pnas.96.18.10242

15. Liebeke, M., Bruford, M.W., Donnelly, R.K., Ebbels, T.M.D., Hao, J., Kille, P., … Bundy, J.G. (2014). Identifying biochemical phenotypic differences between cryptic species. Biology Letters, 10(9), 20140615. https://doi.org/10.1098/rsbl.2014.0615

16. Mischis, C.C. (2007). Catálogo de las lombrices de tierra de la Argentina (Annelida, Oligocheta). En: Brown G.G. & Fragoso C. (eds.), Minhocas: biodiversidade e ecología na America Latina (pp. 241-246). Embrapa So. Londrina, Brasil.

17. Morais, P., & Reichard, M. (2018). Cryptic invasions: A review. Science of the Total Environment 613, 1438-1448. https://doi.org/10.1016/j.scitotenv.2017.06.133

18. Omodeo, P., Rota, E., & Baha, M. (2005). The megadrile fauna (Annelida: Oligochaeta) of Maghreb: a biogeographical and ecological characterization. Pedobiologia, 47(5–6), 458–465. https://doi.org/10.1078/0031-4056-00213

19. Plisko, J. (2001). Notes on the occurrence of the introduced earthworm Pontoscolex corethrurus (Müller, 1857) in South Africa (Oligochaeta: Glossoscolecidae ). African Invertebrates, 42(1), 323–334. https://hdl.handle.net/10520/EJC84473

20. Pyšek, P. , Jarošík, V. , Chytrý, M. , Kropáč, Z. , Tichý, L. and Wild, J. (2005). Alien plants in temperate weed communities: Prehistoric and recent invaders occupy different habitats. Ecology, 86(3), 772–785. https://doi.org/10.1890/04-0012

21. Reynolds, J.W., & Jones, A.G. (2006). The earthworms (Oligochaeta: Acanthodrilidae, Glossoscolecidae, and Lumbricidae) of the Falkland Islands. South Atlantic Ocean. Megadrilogica 10(10), 75-86. 

22. Rota, E., & de Jong, Y. (2015). Fauna Europaea: Annelida - Terrestrial Oligochaeta (Enchytraeidae and Megadrili), Aphanoneura and Polychaeta. Biodiversity Data Journal, 3, e5737. https://doi.org/10.3897/bdj.3.e5737

23. Sherlock, E., & Carpenter, D. (2009). An updated earthworm list for the British Isles and two new “exotic” species to Britain from Kew Gardens. European Journal of Soil Biology, 45(5-6), 431–435. https://doi.org/10.1016/j.ejsobi.2009.07.002

24. Singh, S., Singh, J., Sharma, A., Vig, A.P., & Ahmed, S. (2018). First Report of the Earthworm Pontoscolex corethrurus (Müller, 1857) from Punjab, India. International Letters of Natural Sciences, 68, 1–8. https://doi.org/10.18052/www.scipress.com/ilns.68.1

25. Statsoft. (1999). STATISTICA for Windows (Computer Program Manual). Statsoft, Inc.

26. Steffen, G.P.K., Steffen, R.B., Bartz, M.L.C., James, S.W., Jacques, R.J.S., Brown, G.G., & Antoniolli, Z.I. (2018). Earthworm diversity in Rio Grande do Sul, Brazil. Zootaxa, 4496(1), 562–575. https://doi.org/10.11646/zootaxa.4496.1.43

27. Subedi, H.P., Saxena, R.M., & Reynolds, J.W. (2018). New record of an earthworm in the family Glossoscolecidae (Annelida: Oligochaeta) from Sikkim, India. Megadrilogica, 23(2), 51–56. 

17

18

19

19

19

21

23

24 

380

415

428

433

446

492

528

550

---

## [Editor Report · Decision Letter 1]

9 Aug 2019

PONE-D-19-15357R1

Pontoscolex corethrurus: a Homeless Invasive Tropical Earthworm?

PLOS ONE

Dear Dr Ortiz-Ceballos,

Thank you for submitting your manuscript to PLOS ONE. After careful consideration, we feel that it has merit but does not fully meet PLOS ONE’s publication criteria as it currently stands. Therefore, we invite you to submit a revised version of the manuscript that addresses the points raised during the review process.

We would appreciate receiving your revised manuscript by Sep 23 2019 11:59PM. To enhance the reproducibility of your results, we recommend that if applicable you deposit your laboratory protocols in protocols.io, where a protocol can be assigned its own identifier (DOI) such that it can be cited independently in the future. For instructions see: http://journals.plos.org/plosone/s/submission-guidelines#loc-laboratory-protocols

We look forward to receiving your revised manuscript.

Kind regards,

Tunira Bhadauria, Ph.D

Academic Editor

PLOS ONE

Additional Editor Comments (if provided):

The authors need to be congratulated for sincerely incorporating the changes and suggestions in the revised manuscript. However there are some minor changes which further need to be incorporated into the text before manuscript can be accepted for the publication

1.Line 261 in revised manuscript spelling of which

2.Line 302 Year of reference 199

3.Line 332 sentence incomplete changes incorporated in text

---

## [Author Response · Author response to Decision Letter 1]

13 Aug 2019

August 12, 2019

PlosOne

Editor-in-Chief

Dear Editor:

Please find enclosed the manuscript titled (Tracked Changues version) Pontoscolex corethrurus: a Homeless Invasive Tropical Earthworm? Below is the rebuttal letter in which we list all the changes (point-by-point) we have made to the manuscript.

Thank you again your comments and assistance.

Sincerely yours

Angel I. Ortiz Ceballos

 

Rebuttal Letter

General remarks

All the corrections indicated referees (Editorial Board Editor) was incorporated on the manuscript. In green, sentences and words eliminated; in red, sentences, words incorporated, and only in the text of the manuscript. 

Specific points:

Discussion

Comments:

…However there are some minor changes which further need to be incorporated into the text before manuscript can be accepted for the publication Page Line

Response: 

We agree with these comments. We include and delete the following information suggested:

Delete (Line 261 in revised manuscript spelling of which):

, wich differ from those in the adjacent forest

Delete (Line 302 Year of reference 199):

(Daane et al. 199)

Include:

(Daane et al. 1999)

Delete (Line 332 sentence incomplete changes incorporated in text):

Based on the results reported here, we conclude that the lower survival and hatching rates of P. corethrurus cocoons (offspring) in the agro-ecological niche of B. pearsei is… 

Include:

Based on the results reported here, we conclude that the invasive tropical earthworm P. corethrurus had lower survival and cocoons hatching rates (offspring) in the agro-ecological niche of the native endogeic earthworm, i.e., a finding…

Include:

Acknowledgments

We thank Mario M. Osorio-Arce, Angel Ramos-Sánchez and Efraín Hernández-Xolocotzi, promoters of agroecology in Mexico. In addition, we are grateful to anomymous reviewers and Diana Pérez-Staples for valuable comments and careful revision of the manuscript. 

12

13

15

15

260

302

331

342

---

## [Editor Report · Decision Letter 2]

28 Aug 2019

Pontoscolex corethrurus: a Homeless Invasive Tropical Earthworm?

PONE-D-19-15357R2

Dear Dr. Ortiz-Ceballos,

We are pleased to inform you that your manuscript has been judged scientifically suitable for publication and will be formally accepted for publication once it complies with all outstanding technical requirements.

With kind regards,

Tunira Bhadauria, Ph.D.

Academic Editor

PLOS ONE

Additional Editor Comments (optional):

i would like to congratulate the authors for having revised the manuscript as per the suggestions and comments and incorporation of the same at appropriate places in the text. I therefore recommend that the manuscript now can be accepted for publication in the journal.
---

## [Editor Report · Acceptance letter]

6 Sep 2019

PONE-D-19-15357R2 

*Pontoscolex corethrurus*: a Homeless Invasive Tropical Earthworm? 

Dear Dr. Ortiz-Ceballos:

I am pleased to inform you that your manuscript has been deemed suitable for publication in PLOS ONE. Congratulations! Your manuscript is now with our production department. 

With kind regards,

on behalf of

Dr. Tunira Bhadauria 

Academic Editor

PLOS ONE